# Digital Media Influence on Adolescents’ Behavior during the COVID-19 Pandemic: Self-Intravenous Injection of *Lactobacilli* Drinking Yogurt

**DOI:** 10.3390/children9081104

**Published:** 2022-07-22

**Authors:** Eunjung Koh, Insu Choi, Seul-A Choi, Yeo-Jin Kang, Hwa-Jin Cho

**Affiliations:** Department of Pediatrics, Chonnam National University Children’s Hospital and Medical School, Gwangju 61706, Korea; mh0360@naver.com (E.K.); tmfdk3934@naver.com (S.-A.C.); yeojinplz@naver.com (Y.-J.K.); chhj98@gmail.com (H.-J.C.)

**Keywords:** *Lactobacillus*, drinking yogurt, bacteremia, digital media, adolescence, self-intravenous injection

## Abstract

During the coronavirus disease 2019 (COVID-19) pandemic, widened physical distance and social deprivation are being replaced with digital media use. The media’s social function has tremendously increased following the COVID-19 era and adolescents living in a media-saturated world are the most vulnerable subjects of all. Social media use can encourage risky behavior in adolescents. Posting videos related to risk behaviors on social media has been linked to jeopardizing not only mental health but also physical safety. Herein, we report a case of a 12-year-old boy who intravenously self-injected *Lactobacilli* drinking yogurt for the purpose of filming a video to post on social media. He was treated with antibiotics based on *Lactobacilli* sepsis that cured him without any complications. In order to reduce the risk-taking behavior of adolescents, social norms need to be straightened up, and the social responsibility of hosts is strongly recommended.

## 1. Introduction

During the coronavirus disease 2019 (COVID-19) pandemic, widened physical distance and social deprivation are being replaced with digital media use [1]. Even before the pandemic, social media platforms shared information, ideas, opinions, messages, images and videos; the media’s social function tremendously increased following the COVID-19 era [1]. In addition to the advantages mentioned above, social media use can encourage risky behavior in the form of posting videos of excessive drinking or self-harm [2,3]. To understand the relationship between social media use and risky behavior, it is important to know the specificity of adolescence development. Adolescence is a period when the influence of parents decreases and the influence of peers becomes important [4]. A previous study demonstrated that adolescents’ decision making is affected simply by the presence of peers [5]. Thus, the significance of social media in adolescence is related to the growing importance of identity exploration, self-expression, friendships, and peer acceptance in this developmental period [6]. Moreover, social media platforms may inform the concept of risky behavior and may give a sense of belonging to their peers who show similar risky behavior [7]. Nowadays, adolescents living in a media-saturated world are able to post self-generated content without much effort and are the most vulnerable subjects of all. Many studies demonstrated a relationship between social media use and concomitantly increasing rates of self-harm, non-suicidal self-injury, and other forms of potential risky behaviors [8]. Herein, we report a case of a 12-year-old boy who intravenously self-injected *Lactobacilli* drinking yogurt for the purpose of filming a video to post on social media. This is a unique case of bizarre risky behavior for the purpose of posting on social media.

## 2. Case Report

A 12-year-old boy was transferred to the emergency department presenting with left arm pain and fever claiming that he had intravenously self-injected *Lactobacilli* drinking yogurt. He showed no signs of mental change nor lethargy. He was hospitalized at another clinic for treatment of tonsillitis using intravenous antibiotics 4 days before admission. While his sore throat and pyrexia improved, he played with another patient’s syringe which was given for nebulizer fluid. With the syringe, he injected 4 mL of *lactobacilli* drinking yogurt, 2 mL of tap water and 5 mL of air several times through a 3-way stopcock of the intravenous line while recording his actions. After 30 min, he complained of myalgia with pyrexia and was transferred to our emergency department. His white blood cell count (WBC) was 3200/μL with 58.6% neutrophil and 39.9% lymphocyte. Platelet count was 195,000/μL with C-reactive protein (CRP) level of 0.74 mg/dL. Venous blood gas analysis, electrolytes and other disseminated intravascular coagulation lab were in the normal range. Chest X-ray showed no sign of cardiomegaly. His Glasgow coma scale was 15, but we started ampicillin/sulbactam and cefotaxime as empirical antibiotics for *Lactobacilli* intravenous injection. Then, 13 h after injection, his fever interval became shorter and follow-up laboratory findings showed an elevated CRP level of 14.36 mg/dL with procalcitonin 20.80 ng/mL, WBC 6200/μL with 75.5% neutrophil and platelet count 117,000/μL. Cardiac echo and laboratory findings of cardiac markers showed no sign of endocarditis or myocarditis. We changed antibiotics to broad spectrum antibiotics, meropenem and vancomycin. Fortunately, his vital signs remained stable and fever subsided after 28 h of drinking yogurt injection.

Cells isolated from initial blood cultures were stained Gram-positive (Figure 1), and were taken from two separate sites. Cultures grew well under anaerobic conditions. Characteristic colonies grew from BHI blood agar, which is shown in Figure 2. We used MALDI-TOF to identify *Lactobacillus paracasei*.

Followed-up blood cultures showed no sign of bacteremia, and CRP level decreased to normal range.

There were possibilities of suicide attempt, adjustment disorder, mischief, drug abuse and intellectual disability. We checked for any possible anxiety related to starting middle school, stress from studying, or unusual mood changes. However, the boy showed no signs of suicide attempt nor major depressive disorder. The boy admitted that he watched digital media and intentionally copied it out of curiosity to brag to his friends by posting such a video of himself.

## 3. Discussion

Adolescence is a period between childhood and adulthood when a developmental transition takes place, with less care for parental influence but more care for importance among peers [4]. Due to the strong need to be accepted by peers, they are prone to conform to the social norms [9]. Social norms theory, suggests that individual behavior can be affected by the perceived behavior of others despite a moral conscience. It can be applied to explain the relationship between social media and hazardous behavior [10]. The COVID-19 era encouraged people to use social media more often. In order to attract more viewers, some users started to post more stimulating and provoking material that had never been posted before. This content was exposed to adolescents without any regulation and social norms expanded to wider boundaries, permitting or even encouraging risk-taking behavior. It becomes a more serious problem because adolescents are prone to take more risks in the presence of peers and when peers stimulate risk-taking, than any other age group [11]. The presence of peers makes their subjective value of rewards arousing, thereby risky behavior is preferred over safe alternatives [12].

Of note, not all types of digital media use have adverse consequences on adolescents’ mental health [1]. This can only be achieved if we publicize the detrimental influence of social media on the mental health of susceptible adolescents and make them realize the adverse mechanisms such as social comparison, fear of missing out, and exposure to negative content, which are more likely to happen during social isolation and confinement due to the pandemic [11].

In this case, the boy survived an infection caused by self-injected *Lactobacilli* drinking yogurt, tap water and a large amount of air with a non-sterile syringe because of his early confession and immediate intervention with broad spectrum antibiotics. Additionally, the immune status of the boy was crucial. *Lactobacilli* are anaerobic Gram-positive rods that are normal flora colonizing in gastrointestinal and genitourinary areas [13]. *Lactobacilli* are normally considered to be contaminants in blood culture, but they have been reported to cause endocarditis, urinary tract infections, meningitis, intra-abdominal infections and bacteremia [14]. Although clinically significant *Lactobacillus* bacteremia was reported from immunocompromised patients with cancer, Crohn’s disease, diabetes mellitus, and recent surgery [14,15], a case of *Lactobacillus endocarditis* in an immunocompetent patient with probiotics use and gingival laceration was also reported [16]. Previous studies have reported that *Lactobacillus* species are resistant to vancomycin, ciprofloxacin, tetracyclin, meropenem, and metronidazole [17]. Combination therapy with penicillin and aminoglycoside was shown to be an effective treatment [18].

Fortunately, our case patient survived from *Lactobacilli* sepsis without any complication. However, this could have led to sudden death if he had not told us what he did, as the cause would remain unknown. There was also a possibility of air embolism in addition to sepsis.

There are many studies examining the relationship between social media use and risky behavior during adolescence. Several studies demonstrated that the more adolescents that are exposed to social media, the more risky behavior they adopt, including substance use, risky sexual behavior, and violent behavior [19,20]. Other studies identified that social media use can also have harmful effects on adolescents such as cyberbullying, sexting, sending embarrassing photos, publicly sharing location, and the sharing of dangerous pranks and games [21,22,23,24,25]. In a recent study, social media revealed to be a means by which adolescents self-reported sharing thoughts and actions of self-harm with others [26]. Therefore, several social media platforms have started to limit the sharing of explicit pictures or messages of self-harm.

In order to reduce the risk-taking behavior of adolescents, social norms need to be straightened up, and the social responsibility of hosts and advertising companies is strongly encouraged. Social media can be used as a platform for prevention or intervention to set social norms straight. Each company has a self-regulatory policy and age restriction for violent or sexual content. Reinforcing restriction can be a solution but it can also violate freedom of speech. Advertising companies sponsor private social media hosts because customization through social media advertising plays a main role on advertisement value [27]. As advertisers seek hosts with many viewers, hosts are eager to attract more subscribers by practicing difficult and eccentric acts that have never been done before, such as eating very spicy food covered in capsaicin, jumping off a cliff, or choking games that resulted in hypoxic seizures [28]. By raising awareness of social media’s huge impact on social norms, advertisement companies should take responsibility for their actions and try to avoid sponsoring hosts that post inappropriate content.

There are increasing concerns that digital media may encourage risky behavior in adolescents. This case demonstrates a relationship between digital media use and its detrimental effect on users. Although there is controversy over whether the use of digital media only has negative effects, clinicians should be aware of such adolescents’ unexpected behavior.

## Figures and Tables

**Figure 1 children-09-01104-f001:**
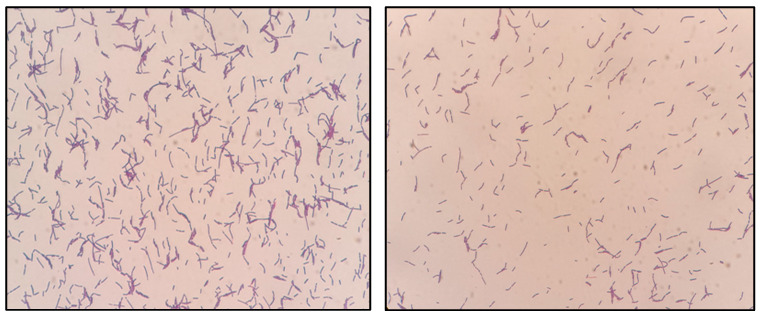
Gram-positive bacilli revealed by Gram staining in blood sample cultures, which were later identified as *Lactobacillus paracasei* by matrix-assisted laser desorption/ionization time-of-flight (MALDI-TOF).

**Figure 2 children-09-01104-f002:**
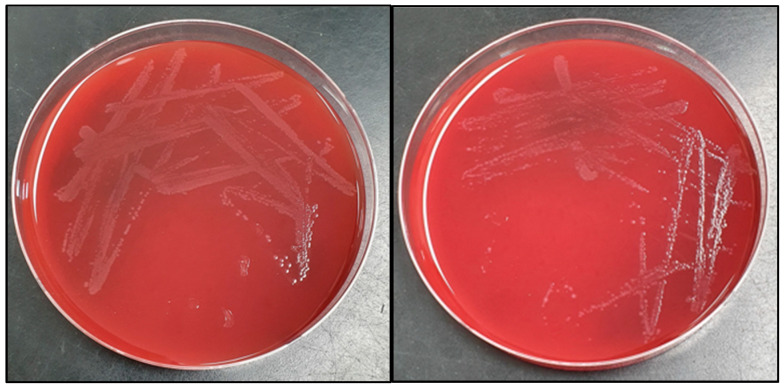
Gray colonies of 1–2 mm diameter of bacteria, observed after incubation of BHI blood agar at 35 °C for 24 h in 5% carbon dioxide gas culture.

## Data Availability

All data and material analyzed in this study are included in this published article.

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
