# Peer review of "Digital Media Influence on Adolescents’ Behavior during the COVID-19 Pandemic: Self-Intravenous Injection of Lactobacilli Drinking Yogurt"

_children, 2022, doi:10.3390/children9081104_

Round 1

Reviewer 1 Report

The authors report an incident during COVID which shows the impact of COVID on children and extreme measures taken by them to get attention due to lack of other activities. 

There is a lot of medical terminology which would be of interest to people from medical profession. The discussion focuses on medical as well as  psychological affects. The discussion on psychological aspects could be expanded along with probable remedies. What could possibly be done to minimize such incidents. 

The manuscript requires minor proof reading.  

Author Response

Thank you very much for your time and we all appreciate your input.

The authors report an incident during COVID which shows the impact of COVID on children and extreme measures taken by them to get attention due to lack of other activities. 

There is a lot of medical terminology which would be of interest to people from medical profession. The discussion focuses on medical as well as  psychological affects. The discussion on psychological aspects could be expanded along with probable remedies. What could possibly be done to minimize such incidents. 

: We totally agree with you. We reinforced our claim with more references and mentioned psychological remedies to minimize such incidents. We added a paragraph in discussion section arguing that ‘Social media can be used as a platform for prevention or intervention to set social norms right.’ and ‘By raising awareness of social media’s big role of affecting social norms, advertisement company should take responsibility for their action and try to avoid sponsoring hosts with inappropriate contents.’ We added references from ‘Evaluating the effect of youtube advertising towards young customers’ purchase intention by Aziza, D. N. et al, ‘The choking game and youtube: A dangerous combination by Linkletter, M. et al to support our claim.

Reviewer 2 Report

General Comments

I can say that the subject of the research is current and interesting. Social media content production and the problems it creates, which accelerated with digitalization before Covid 19, have been the subject of a lot of research recently. For this reason, the subject of the research will be current and important in terms of raising awareness for the readers.

Although the subject is beautiful, it is seen that there were obvious technical mistakes during the writing phase of the article. There are serious errors in the technical writing technique of the manuscript. For this reason, it is necessary to review the article in accordance with the writing and make the necessary adjustments correctly.

The biggest shortcoming was that the abstract section was like a paragraph of the introduction and did not provide information about the general design of the study.

The introduction was written as a general comment. No sentence of the text contained references, which was an important omission. In addition, articles containing research results in the literature were not included.

The introduction part of the discussion part was in the form of the introduction part of the article. In this section, it will be more appropriate for the writing of the section to discuss the results by establishing a relationship with the literature.

Minör revision

Manuscript should be made suitable for article writing. In this regard, it is appropriate to make the following suggestions.

-          The abstract should reflect the template of the article when read. The abstract should contain information about the introduction, method, and discussion sections. This part should be edited.

-          The introduction is written without reference to the literature. It should be rewritten with reference from the literature. Information and results of similar studies should be presented.

-          In the discussion section, the discussion of the results with similar studies in the literature was insufficient. The discussion is written somewhat like an introduction. These parts should be reviewed again and necessary additions should be made.

Author Response

Thank you very much for your time and we all appreciate your input.

Manuscript should be made suitable for article writing. In this regard, it is appropriate to make the following suggestions.

  1. The abstract should reflect the template of the article when read. The abstract should contain information about the introduction, method, and discussion sections. This part should be edited.
  2. The introduction is written without reference to the literature. It should be rewritten with reference from the literature. Information and results of similar studies should be presented.
  3. In the discussion section, the discussion of the results with similar studies in the literature was insufficient. The discussion is written somewhat like an introduction. These parts should be reviewed again and necessary additions should be made.

: We edited abstract in introduction, method, and discussion format. We added ‘Social media use can encourage risky behavior to adolescents. Social media platforms may inform the concept of risky behavior and may give a sense of belonging to the peers who act similar risky behaviors. Posting videos related to risk behaviors on social media has been linked to jeopardizing not only mental health but also physical safety’ for abstract’s introduction part and ‘He was treated with antibiotics based on sepsis and Lactobacilli sepsis was cured without any complication. In order to reduce the risk taking behavior of adolescents, social norms need to be straightened up and social responsibility of hosts and advertising company is strongly recommended.’ for discussion part.

We also have reinforced introduction part with more reference with literatures. ‘Real use or “real cool”: Adolescents speak out about displayed alcohol references on social networking websites, by Moreno, M. A. et al’, ‘Where do youth learn about suicides on the internet, and what influence does this have on suicidal ideation? By Dunlop, S. M. et al’, ‘Is adolescence a sensitive period for sociocultural processing Blakemore, S.-J.et al’.’ Is adolescence a sensitive period for sociocultural processing. By Mills, K. L et al, etc. We elaborated relationship between social medial use and risky behavior and characters of adolescence development to assert the importance of our case. 

We reviewed discussion part again and we added results of similar studies in other literature such as ‘Bullying victimization, social network usage, and delinquent coping in a sample of urban youth: Examining the predictions of general strain theory. By Baker, T. et al’, ‘Social media use and risky behaviors in adolescents: A meta-analysis by Vannucci, A. et al’, ‘Risky behaviors and social networking sites: How is youtube influencing our youth? By Ahem, N.R. et al.’ etc to reinforced our claim with more references and discussed severeal psychological remedies to minimize such incidents.

We apologize for not having references marked. We have checked it several times to not to make the same mistake.